# Acid and Neutral Sphingomyelinase Behavior in Radiation-Induced Liver Pyroptosis and in the Protective/Preventive Role of rMnSOD

**DOI:** 10.3390/ijms21093281

**Published:** 2020-05-06

**Authors:** Samuela Cataldi, Antonella Borrelli, Maria Rachele Ceccarini, Irina Nakashidze, Michela Codini, Oleg Belov, Alexander Ivanov, Eugene Krasavin, Ivana Ferri, Carmela Conte, Federica Filomena Patria, Tommaso Beccari, Aldo Mancini, Francesco Curcio, Francesco Saverio Ambesi-Impiombato, Elisabetta Albi

**Affiliations:** 1Department of Pharmaceutical Sciences, University of Perugia, 06126 Perugia, Italy; samuelacataldi@libero.it (S.C.); chele@hotmail.it (M.R.C.); irinanakashidze@yahoo.com (I.N.); michela.codini@unipg.it (M.C.); carmela.conte@unipg.it (C.C.); patriafederica@gmail.com (F.F.P.); tommaso.beccari@unipg.it (T.B.); 2MolecularBiology and Viral Oncology Unit, Istituto Nazionale Tumori IRCCS "Fondazione G. Pascale", 80131 Napoli, Italy; a.borrelli@istitutotumori.na.it; 3Laboratory of Radiation Biology, Joint Institute for Nuclear Research, 141980 Dubna, Russia; dem@jinr.ru (O.B.); a1931192@mail.ru (A.I.); krasavin@jinr.ru (E.K.); 4Division of Pathological Anatomy and Histology, Department of Experimental Medicine, School of Medicine and Surgery, University of Perugia, 06126 Perugia, Italy; ivanaferri@gmail.com; 5Laedhexa Biotechnologies Inc., San Francisco, CA 94130, USA; aldo_mancini@tiscali.it; 6Dipartimento di Area Medica, University of Udine, 33100 Udine, Italy; francesco.curcio@uniud.it (F.C.); saverio.ambesi@uniud.it (F.S.A.-I.)

**Keywords:** acid sphingomyelinase, neutral sphingomyelinase, radiation, SOD, liver

## Abstract

Sphingomyelins (SMs) are a class of relevant bioactive molecules that act as key modulators of different cellular processes, such as growth arrest, exosome formation, and the inflammatory response influenced by many environmental conditions, leading to pyroptosis, a form of programmed cell death due to Caspase-1 involvement. To study liver pyroptosis and hepatic SM metabolism via both lysosomal acid SMase (aSMase) and endoplasmic reticulum/nucleus neutral SMase (nSMase) during the exposure of mice to radiation and to ascertain if this process can be modulated by protective molecules, we used an experimental design (previously used by us) to evaluate the effects of both ionizing radiation and a specific protective molecule (rMnSOD) in the brain in collaboration with the Joint Institute for Nuclear Research, Dubna (Russia). As shown by the Caspase-1 immunostaining of the liver sections, the radiation resulted in the loss of the normal cell structure alongside a progressive and dose-dependent increase of the labelling, treatment, and pretreatment with rMnSOD, which had a significant protective effect on the livers. SM metabolic analyses, performed on aSMase and nSMase gene expression, as well as protein content and activity, proved that rMnSOD was able to significantly reduce radiation-induced damage by playing both a protective role via aSMase and a preventive role via nSMase.

## 1. Introduction

Sphingomyelins (SMs) are a class of bioactive lipid molecules that act as key modulators of different pathophysiologic processes, including cell growth, cell death, autophagy, stress, inflammatory responses, and cancer [1]. Sphingomyelinases (SMase) are a family of key enzymes in SM metabolism that generate the ceramide and phosphorylcholine headgroups. From a cellular perspective, the existence of the isoenzyme’s multiplicity has functional reasons. Our improved understanding of these isoenzymes has provided information on the different roles of SMs [2]. SMases are named based on their optimal pH activity as an acid, neutral, or alkaline SMase, with different locations and functions inside the cells [3]. Alkaline SMase (Alk-SMase) shares no structural similarities with the other two SMases; it belongs to the ecto-nucleotide pyrophosphatase phosphodiesterase (NPP) family and is located in the mucosal membrane of the intestinal tract [4]. Acid SMase (aSMase) and neutral SMases (nSMases) have organelle-specific activities and distinct regulatory mechanisms. Thus, the aSMase isoform is located in the lysosome and is involved in apoptosis signaling [5]. Moreover, four nSMase isoforms are located in the inner and outer leaflet of the plasma membrane, endoplasmic reticulum, mitochondria, and nucleus [6]. Enzyme localization influences the biologically relevant activation mechanisms. Moreover, the concentration of the SM and ceramide is organelle dependent. nSMase isoforms have been identified on the basis of four genes that were cloned or purified: nSMase1 (gene name = SMPD2), nSMase2 (SMPD3), nSMase3 (SMPD4), and MA-nSMase (mitochondrial-associated nSMase) (SMPD5) [6]. nSMase1 is located in the reticulum endoplasmic/Golgi apparatus [7], as well as in the cell nucleus [8]. It is activated in response to stress by inducing apoptosis [9] and also suppresses hepatocellular carcinoma [10]. nSMase2 is specific to the inner leaflet of the plasma membrane and is involved in many cell responses, such as cell growth arrest, exosome formation, and inflammatory response [11]. nSMase3 is located in the endoplasmic reticulum and is involved in TNF-α mediated signaling, tumorigenesis [6], and cellular stress response [12]. MA-nSMase, which has sequence homology with nSMase2 and zebrafish mitochondrial N-SMase [13], was identified only in 2010 [14]. 

It has been demonstrated that SM metabolism is finely modulated in the liver [15], an active metabolic organ influenced by many environmental conditions, including radiation [16]. Irradiation induces DNA repair activities after DNA damage in hepatocytes (and inflammatory reactions in other cell types [16]) by leading to pyroptosis, a form of programmed cell death due to Caspase1 involvement [17], which is characterized by membrane rupture, pore formation, and the release of pro-inflammatory cytokines [18]. Therefore, pyroptosis is important as a final event of radiation-induced damage. 

The effect of ionizing radiation on liver SM metabolism has not yet been clarified. It was previously shown that in thyroid cells, ionizing radiation exposure induces proapoptotic signals via ceramide production from the SMs [19]. Moreover, proton beams move quiescent thyroid cells towards a proapoptotic state and proliferating thyroid cells towards an initial apoptotic state by altering the nuclear SM-metabolism [20]. In the same experimental model, ultraviolet radiation enriched the ceramide pool due to the ability of both aSMase and nSMase to induce apoptosis [21].

Thus, the apoptotic process requires the action of both aSMase and nSMase (especially nSMase1 [9]), but whether there is cooperation between the two enzymes and whether they behave differently in relation to the same apoptotic stimulus has not yet been investigated. To study what happens to the metabolism of hepatic SM via both the lysosomal and endoplasmic reticulum/nucleus SMase during irradiation, and to ascertain if this process can be modulated by protective molecules, we used the same experimental model previously applied to evaluate the effects of both ionizing radiation and a specific protective molecule in the brain [22]. This research originated from a collaborative project among Italian research groups and the Joint Institute for Nuclear Research, Dubna (Russia), in which mice were exposed to a set of minor γ radiation, neutrons, and a spectrum of neutrons, simulating the radiation levels that cosmonauts are exposed to during deep-space long-term missions. In the brain, radiation was shown to deconstruct neurofilaments in a dose-dependent manner with an increase of the nSMase3 gene and protein expression. Human recombinant manganese superoxide dismutase (rMnSOD), which has a protective and preventive role on brain damage, strongly increased nSMase expression and activity [22]. Since ionizing radiation induces oxidative stress, and rMnSOD has specific antioxidant and anti-free radical activity, in the previous study, we analyzed the behavior of the nSMase3 that is stimulated in the cellular stress response [12]. Here, we analyzed the behaviors of two enzymes that are involved in apoptosis, aSMase and nSMase1, which are capable of degrading MS in two different cellular districts. We performed our study on the liver, which is an organ that actively reacts to radiation though parenchymal cells or hepatocytes in the G0 phase of the cell cycle, thereby regulating the metabolism of many factors, including lipids [16]. Thus, considering our extensive experience with the role of SM metabolism in liver proliferation and apoptosis [23,24], we evaluated aSMase and nSMase in relation to radiation-induced pyroptosis and their response to rMnSOD treatment. The present paper reports the results of an observational study of a novel experiment that could be useful to the scientific community as the basis for future work. 

## 2. Results

### 2.1. Ionizing Radiation Effects on the Liver and the Role of rMnSOD 

The microscopy analysis, performed on histological microsections of the control livers (CTR, rMnSOD untreated, and un-irradiated mice) subjected to Caspase-1 immunostaining, showed normal cells with a very low percentage of labelling (Figure 1a,b). Radiation induced a loss of the normal cell structure alongside a progressive and dose-dependent increase in labelling. The images provide evidence of a significant increase in irregular cellular shapes and membrane ruptures compared to the CTR sample (Figure 1a). The quantification of Caspase-1 showed an increase of 2.8, 3.9, and 5.1 times in the labelling (0.25 Gy, 0.5 Gy, and 1.0 Gy, respectively) compared to the CTR samples. The treatment with rMnSOD alone did not induce significant variation with respect to the CTR but reduced the effect of radiation when administered preventively (see Materials and Methods). The labelling increased by 2.0, 2.5, and 4.1 times with 0.25 Gy, 0.5 Gy, and 1.0 Gy, respectively, compared to the CTR samples. Pretreatment with rMnSOD for preventive purposes had an even greater effect, as labelling increased by only 3.4 times among the mice receiving 1.0 Gy of radiation (Figure 1a,b).

### 2.2. Changes of Sphingomyelin Metabolism

Our previous studies indicated that radiation targets SMase in the thyroid [20,21] and brain [22]. As there are two SMases involved in the apoptotic process (lysosomal aSMase and endoplasmic reticulum/nucleus nSMase1), we defined their behavior in the liver, where radiation upregulated Caspase-1, thereby triggering pyroptosis. We first measured SMPD1 (coding for aSMase) and SMPD2 (coding for nSMase1) gene expression in livers from a) CTR mice, b) rMnSOD treated mice, and un-irradiated mice; c) 0.25 Gy, 0.5 Gy, and 1.0 Gy irradiated mice and mice untreated with rMnSOD; d) 0.25 Gy, 0.5 Gy, and 1.0 Gy irradiated and rMnSOD treated mice; and e) mice pretreated with rMnSOD and irradiated with 1.0 Gy radiation (Figure 2). The results show that SMPD1 was overexpressed by 2.23 + 0.34, 7.05 + 0.42, and 14.1 + 1.47 times with 0.25 Gy, 0.5 Gy, and 1.0 Gy radiation, respectively. The gene expression of SMPD1 did not vary when treated with rMnSOD alone. Treatment with rMnSOD limited the effects of radiation among the irradiated mice and reduced the effects of 0.25 Gy by 19.3%, that of 0.5 Gy by 62%, and that of 1.0 Gy by 75%. The use of rMnSOD as a method of damage prevention was less effective. Notably, the effect of 1.0 Gy radiation was reduced by 44%. These results suggest that rMnSOD plays a limited role in controlling SMPD1 expression when it is used as a preventive molecule for radiation-induced damage, while also being an effective protective molecule. 

We then tested the expression of the SMPD2 gene coding for nSMase1. Its variations under radiation treatment, with or without rMnSOD, were very low (Figure 2). 

To date, the changes of both aSMase and nSMase1 proteins induced by increasing radiation doses and/or rMnSOD have not been analyzed. Thus, we determined if the changes caused by radiation at the genetic level were consistent with protein variation. Using aSMase and nSMase1 specific antibodies, we were able to measure the level of proteins relative to the CTR samples (Figure 3a). The results related to aSMase, normalized for β-tubulin, showed that the enzyme was reduced by 18%, 52%, and 34% with 0.25 Gy, 0.5 Gy, and 1.0 Gy, respectively (Figure 3b). The reduction of protein levels despite increased gene overexpression strongly suggests an increased degradation of the enzyme. Treatment with rMnSOD alone caused a significant reduction in protein compared to the CTR, even when gene expression did not change, possibly because rMnSOD slowed the synthesis of the enzyme due to its proapoptotic role. This effect remained evident with 0.25 Gy and 0.5 Gy of radiation, but very high radiation (1.0 Gy) strongly limited the action of the rMnSOD for both protective and preventive purposes (Figure 3a,b). Therefore, a high radiation dose would hinder the action of rMnSOD.

Conversely, nSMase1 content did not change with irradiation (Figure 3a,b). Surprisingly, rMnSOD alone strongly reduced the nSMase1 form with an apparent 48 kDa molecular weight while inducing the formation of a band with an apparent molecular weight of approximately 28 kDa. Radiation hindered the strong protein reduction obtained via rMnSOD with milder action at a 0.25 Gy dose and a much more intense effect at 0.5 Gy and 1.0 Gy doses (Figure 3a,b). 

To investigate whether ionizing radiation could affect SMase activity, we employed a specific assay kit, as reported in the Materials and Methods. This is a powerful analytical technique that measures the enzymatic activity of SMases, distinguishing aSMase from nSMase by their pH values. We observed that aSMase activity did not change significantly with radiation in both the absence and presence of rMnSOD in the CTR samples (Figure 4). Interestingly, the activity of the nSMase pool was very high in the CTR and was inhibited by radiation. rMnSOD alone increased the activity by 1.65 times in comparison with CTR, but its ability to stimulate this activity was reduced by radiation in a dose-dependent manner. Pretreatment with rMnSOD was able to strongly increase nSMase activity, with values 2× higher than those of CTR.

## 3. Discussion

Caspase-1 is recognized as a molecule involved in the canonical signaling pathway that induces pyroptosis [25,26], a specific type of programmed cell death [27] characterized by membrane rupture and pore formation. Since pyroptosis occurs in different liver diseases [28] and is stimulated by radiation [29], we studied the damage induced in the liver with ionizing radiation treatment in mice, observing histological microsections of liver tissue fixed and stained with anti-Caspase1 antibodies. The specific experimental model used for the present study permitted us to simultaneously investigate the effect of ionizing radiation and the protective/preventive effect of rMnSOD against radiation-induced liver damage. This was a unique opportunity under a collaborative project among Italian research groups and the Joint Institute for Nuclear Research, Dubna (Russia) [22]. Our results provide evidence that both prominent signs of pyroptosis (i.e., the loss of cell membrane integrity and the overexpression of Caspase-1) are induced by radiation treatment in a dose-dependent manner. These effects were limited by rMnSOD administered as a protective agent. Therefore, rMnSOD reduced radiation-induced pyroptosis. If the rMnSOD was administered before irradiation as a preventive agent, its effect was less marked. Thus, the data presented here reveal that rMnSOD played an important role in protection from radiation-induced damage. In our previous study, conducted on the same experimental model, we clearly demonstrated that rMnSOD is able to limit radiation-induced damage to the brain by protecting the brain from the destructuring of neurofilaments with the involvement of nSMase [22]. These results induced us to investigate the behavior of aSMase and nSMase in the liver in association with the expression of Caspase-1. 

Interestingly, SMPD1 (the gene coding for aSMase) was overexpressed following radiation treatment in a dose dependent manner. Considering the important effect of aSMase in apoptosis [30], we expected an upregulation of the enzyme in the liver featuring the characteristics of pyroptosis. Surprisingly, the content of the aSMase protein was lower than that of the control sample, and the enzyme activity remained at low control values. We hypothesized that radiation induced SMPD1 gene overexpression would lead to the synthesis of aSMase, which, in turn, would stimulate the synthesis of Caspase-1. The latter, triggering cell damage, might be responsible for lysosomal rupture with a consequent loss of the aSMase protein. A time-dependent study could have clarified this phenomenon, but such a study was not compatible with the irradiation windows available to us in Dubna when the above results were collected. In support of our data however, recent research has reported the role of aSMase in activating Caspase-1 via the inflammasome, which mediates radiation-induced pyroptosis [31] and the consequent rupture of the lysosome [32]. In this scenario, the use of rMnSOD as a molecule administered alone showed a strong reduction of the aSMasi protein, even though its gene expression did not change. We hypothesize that rMnSOD slowed the synthesis of the enzyme because of its proapoptotic role. Thus, rMnSOD was able to reduce the synthesis of aSMase in the presence of low and medium radiation doses but not at a higher dosage. However, the values of the protein content at all radiation doses always remained lower than those of the control samples in accordance with the mechanism hypothesized above. In case of significant aSMase protein damage due to pyroptosis, rMnSOD might protect the aSMase protein, thereby reducing cellular and tissue damage. Indeed, despite the reduced SMPD1 overexpression following 1.0 Gy irradiation + rMnSOD and rMnSOD + 1.0 Gy compared to the 1.0 Gy irradiated sample, aSMase protein expression was not significantly higher in the 1.0 Gy irradiation + rMnSOD group but instead significantly higher in the rMnSOD + 1.0 Gy group, thereby indicating reduced aSMase damage

The relevance of nSMase isoforms in cellular physiopathology, underscored by the role of nSMase1 in apoptosis, led us to investigate the behavior of this isoenzyme in ionizing radiation-induced pyrotosis. Our results demonstrate that nSMase1 gene and protein expression did not change with radiation. rMnSOD alone strongly reduced nSMase1 with its apparent molecular weight present in all other samples (48 kDa). Unexpectedly, a much lower molecular weight band (28 kDa) appeared in our Western Blotting. Interestingly, a specific isoform of nSMase1 (isoform) with an apparent molecular weight of 28 kDa, inhibited by reduced glutathione, was recently reported [33]. In the presence of radiation, the reduction of nSMase1 by rMnSOD was attenuated, indicating that MnSOD was unable to reduce the synthesis of nSMase1 at medium / high radiation doses. As reported in the results, the method used for nSMase activity was not specific for the nSMase1 isoform, but the results indicated the activity of the nSMase pool. We thus carried out experiments to evaluate whether nSMase activity might change in liver pyroptosis, as in previously reported midbrain radiation-induced damage [22]. Interestingly, radiation inhibited nSMase activity more than the control samples, while rMnSOD alone increased strongly, but this effect was reduced by radiation in a dose-dependent manner when used as a protective agent. Conversely, the use of rMnSOD as a preventive agent strongly increased nSMase activity. Since nSMase1 mainly resides in the nuclear matrix, and to a lesser extent in the endoplasmic reticulum [8], rMnSOD was likely able to act strongly on the plasma membrane nSMase but unable to act significantly at a nuclear level. 

These results contrast with previous studies showing the stimulation of nSMase with ionizing radiation (γ rays) [19], protons [20], and ultraviolet radiation [21]. The explanation for this result is twofold. 1) First, there is the issue of tissue-specificity. The experiments in previous studies were carried out on normal and cancer thyroid cells, while our data focus on the liver. 2) Also important are the dose and type of radiation. Sautin et al. [19] used gamma radiation at a dose of 2–5 Gy, while we used a set of minor γ radiation and neutrons simulating ionizing radiation during space flight at doses of 0.25 Gy, 0.5 Gy, and 1.0 Gy. There are no data in the literature that compare the amount of ROS production in response to the different types of radiation in the liver. Therefore, it is very difficult to establish if the ROS–SMase relationship is also dependent on tissue specificity and / or radiation dose.

The results of the work show a lack of correlation between gene expression, protein expression, and the activities of both the aSMase and nSMase enzymes. We believe this is due to the much more complex in vivo experimental model than the in vitro system, especially for lipid metabolism. Indeed, the in vivo metabolism of lipids, including sphingolipids, is very rapid and influenced by hormones and various metabolic interactions with the production of molecules that can act as stimulators or inhibitors in response to stress conditions, induced by different factors such as radiation or drug treatments. The actions of these molecules can take place at the gene level or during the long process of protein synthesis; conversely, these actions can directly influence the activity of an enzyme. In this specific experiment, we do not know the exact mechanisms that were activated. We have images of the results and can only make assumptions relative to the literature data. 

In conclusion, we currently have no reliable evidence on the direct effect of radiation and rMnSOD on SMases but we have data indicating their variations after different treatments during liver pyroptosis. The possibility of an indirect effect with the involvement of other signal molecules cannot be excluded.

## 4. Materials and Methods

### 4.1. Chemicals

The rMnSOD protein was provided by the Molecular Biology and Viral Oncology Unit, Department of Experimental Oncology, “Istituto Nazionale Tumori Fondazione G. Pascale”—IRCCS, Naples, Italy [34]. Anti-aSMase, anti-nSMase1, anti-Caspase1, and anti-β-tubulin were obtained from Abcam (Cambridge, UK). Horseradish peroxidase-conjugated goat anti-rabbit secondary antibodies were obtained from Santa Cruz. The TaqMan SNP Genotyping Assay and Reverse Transcription kit were obtained from Applied Biosystems (Foster City, CA, USA). The RNAqueous®-4PCR kit was obtained from Ambion Inc. (Austin, Texas, USA). The SDS-PAGE molecular weight standards were purchased from Bio-Rad Laboratories (Hercules, CA, USA). Chemiluminescence kits were purchased from Amersham (Rainham, Essex, UK). 

### 4.2. Experimental Model

The experimental model was the same previously reported [22]. Animals: 54 female mice weighting approximately 25–30 g were obtained from the Laboratory Animal Nursery of the Russian Academy of Sciences (Pushchino, Russia). Mice were adapted to the vivarium at the “Joint Institute for Nuclear Research (JINR)” in Dubna over a period of 10 days. Then, they were divided into 9 cages with 6 mice each, each receiving standard briquetted fodder and water ad libitum. All procedures were performed according to the Russian Guidelines for the Care and Use of Experimental Animals and Bioethics Instructions (Order of the USSR Ministry of Health No. 755 12.08.1987). The mice were divided into groups and numbered with progressive numbers: a) numbers 1,3,4, and 5 were treated with daily subcutaneous injections of sterile PBS solution for 7 days from the day of irradiation; b) mice 2, 6, 7, and 8 were treated with daily subcutaneous injections of rMnSOD in sterile PBS for 7 days from the day of irradiation; c) 9 received a total of 10 injections; they were pretreated with rMnSOD in sterile PBS for 3 days prior to irradiation and then for 7 days from the day of irradiation. All animals were irradiated at the JINR; they were exposed to a set of minor γ radiation and neutrons from a Phasatron with high Relative Biological Effectiveness (RBE) and a spectrum of neutrons to simulate space flight exposure. Animals in groups 3 and 6 were exposed to a dose of 0.25 Gy, those in groups 4 and 7 were exposed to a dose of 0.50 Gy, and those in groups 5 and 8 were exposed to a dose of 1.00 Gy. Mice in groups 1 (mock-treated with PBS) and 2 (rMnSOD-treated) were not exposed to radiation and were considered a biological control. At the end of the experiment, all mice were beheaded and had their livers immediately frozen. 

### 4.3. Immunohistochemical Analysis

Three livers from each group were fixed in 4% neutral phosphate-buffered formaldehyde solution for 24 h and dropped in a specific orientation into paraffin. Immunohistochemical analyses were performed as previously reported [35] using the anti-Caspase-1 antibody. A bond Dewax solution was used for removal of paraffin from tissue sections before rehydration and immunostaining was performed in the Bond automated system (Leica Biosystems Newcastle Ltd, UK). The observations were performed using inverted microscopy EUROMEX FE 2935 (ED Amhem, The Netherlands) equipped with a CMEX 5000 camera system (20× magnification). The analysis of labelling was performed using the ImageFocus software.

### 4.4. Reverse Transcription Quantitative PCR (RTqPCR)

Total RNA was extracted from the livers using an RNAqueous-4PCR kit. Its integrity was evaluated, and the cDNA was synthesized as previously reported [35]. RTqPCR was performed using a TaqMan®Gene Expression Master Mix and a 7500 RT-PCR instrument (Applied Biosystems), targeting genes of SM phosphodiesterase 1 (SMPD1, Hs03679347_g1) and SM phosphodiesterase 2 (SMPD2, Hs04187047_g1) genes. The mRNA expression levels were then normalized to those of the glyceraldehyde-3-phosphate dehydrogenase (GAPDH, Hs99999905_m1) housekeeping gene (Thermo Fisher Scientific, MA, USA). The relative mRNA expression levels were calculated as 2−ΔΔCt and compared to the results of the treated samples with the control and/or with those of the untreated ones [35].

### 4.5. Western Blotting

Protein concentrations were analyzed and electrophoresis was performed as previously reported [36]. Proteins were transferred onto a 0.45 μm cellulose nitrate strip membrane (Sartorius Stedim Biotech S.A.) in a transfer buffer for 1 h at 100 V at 4 °C. Membranes were blocked with 5% (w/v) non-fat dry milk in PBS, pH7.5, for 1 h at room temperature. The blot was incubated overnight at 4 °C with the specific antibodies, anti-aSMase and anti nSMase1 (1:1000), and then treated with horseradish peroxidase-conjugated goat anti-rabbit secondary antibodies (1:5000). A Super Signal West Pico Chemiluminescent Substrate (ThermoFisher Scientific) was used to detect the chemiluminescent (ECL) HRP substrate. The apparent molecular weights of the proteins were calculated in reference to the migration rate of the molecular size standards. The area density of the bands was evaluated by densitometric scanning and analyzed using Scion Image.

### 4.6. aSMase and nSMasi Activity Assay

The aSMase and nSMase activity was assayed according to Conte et al. [37]. Liver homogenates were suspended in 0.1% NP-40 detergent in PBS, sonicated for 30 s on ice at 20 watt, kept on ice for 30 min, and centrifuged at 16,000× g for 10 min. The supernatants were then used for the aSMase and nSMase assay. The enzyme activity was assayed in 60 µg proteins/10 µL Tris-MgCl2, pH 5.0, for aSMase and 7.4 for nSMase using an Amplex Red Sphingomyelinase assay kit (Invitrogen, Monza, Italy), according to the manufacturer’s instructions. The fluorescence was measured with a FLUOstar Optima fluorimeter (BMG Labtech, Germany) using a filter set with a 360 nm excitation and 460 nm emissions.

### 4.7. Statistical Analysis

Data were expressed as the means ± SD of three livers, and their significance was checked by an ANOVA test. Significance: (a) * *p* < 0.05 versus the control sample (CTR); (b) ^§^
*p* < 0.05 rMnSOD treated and irradiated samples versus irradiated samples; (c)^^^
*p* < 0.05 the pretreated and 1.0 Gy irradiated sample versus the 1.0 Gy irradiated and rMnSOD treated sample.

## Figures and Tables

**Figure 1 ijms-21-03281-f001:**
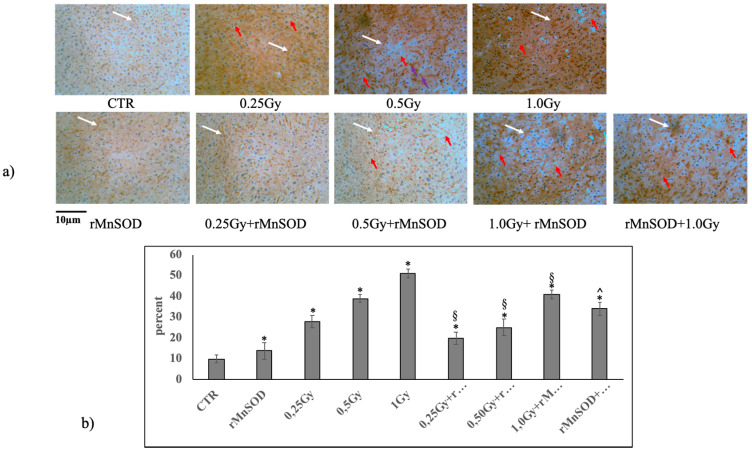
Mouse liver after irradiation with or without protective or preventive rMnSOD treatment (**a**) representative liver histology by Caspase-1 immunohistochemical staining. CTR, control mice; rMnSOD, mice treated with human recombinant manganese superoxide dismutase; 0.25 Gy, 0.5 Gy, and 1.0 Gy, mice exposed to increasing radiation doses; 0.25 Gy + rMnSOD, 0.5 Gy + rMnSOD, and 1.0 Gy + rMnSOD, mice exposed to increasing radiation doses and treated with rMnSOD (protective role of rMnSOD); rMnSOD+1.0Gy, mice pretreated with rMnSOD and exposed to 1.0 Gy radiation (preventive role of rMnSOD). Images are representative of 3 similar images from each group of mice (20× magnification). White arrows indicate positive caspase labelling; red arrows indicate cell membrane ruptures. (**b**) Quantification of Caspase-1 staining was performed using the ImageFocus software. Positive staining is indicated as low (+), medium (++), or high (+++). Only high positive staining was considered and was measured as a percentage of the total area. Data represent the mean + S.D. of three livers for each group. Significance, * *p* < 0.05 with respect to the CTR, ^§^
*p* < 0.05 with respect to the irradiated samples, ^^^
*p* < 0.05 with respect to 1.0 Gy + rMnSOD.

**Figure 2 ijms-21-03281-f002:**
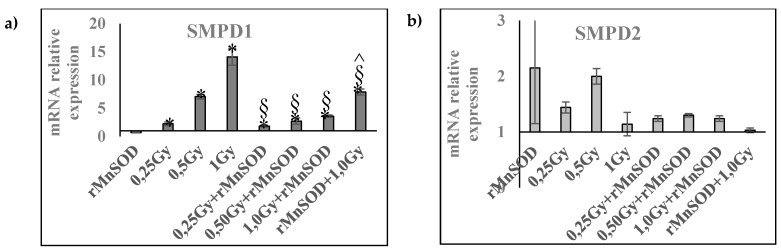
Effect of radiation and rMnSOD on SMPD1 and SMPD2 gene expression in the liver. SMPD1 and SMPD2 gene expression evaluated by RTqPCR as reported in the “Materials and Methods” section. Liver from mice treated with increasing doses of radiation with or without rMnSOS. (**a**) SMPD1 (**b**) SMPD2. Data are expressed as the mean + SD of three liver samples, each carried out in triplicate. Significance: (a) * *p* < 0.05 versus the control sample (CTR); (b) ^§^
*p* < 0.05 rMnSOD treated and irradiated samples versus the irradiated samples; (c)^^^
*p* < 0.05 pretreated and 1.0 Gy irradiated sample versus 1.0 Gy irradiated and rMnSOD treated samples. CTR, control mice; rMnSOD, mice treated with human recombinant manganese superoxide dismutase; 0.25 Gy, 0.5 Gy, and 1.0 Gy, mice exposed to increasing radiation doses; 0.25 Gy + rMnSOD, 0.5 Gy + rMnSOD, and 1.0 Gy + rMnSOD, mice exposed to increasing radiation doses and treated with rMnSOD (protective role of rMnSOD); rMnSOD + 1.0 Gy, mice pretreated with rMnSOD and exposed to 1.0 Gy radiation (preventive role of rMnSOD).

**Figure 3 ijms-21-03281-f003:**
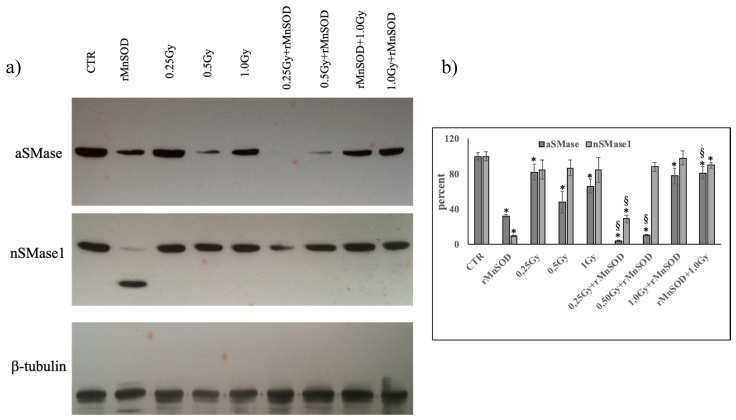
Effect of radiation and rMnSOD on the aSMase and nSMase1 protein level in the liver. The aSMase and nSMase1 level evaluated by Western Blotting, as reported in the “Materials and Methods” section. Liver from the mice treated with increasing doses of radiation with or without rMnSOS. (**a**) western blotting panel; (**b**) densitometric analysis performed using the ImageFocus software. Data are expressed as a percentage with respect to the control sample and represent the mean + SD of three liver samples, each carried out in triplicate. Significance: (a) * *p* < 0.05 versus the control sample (CTR); (b) ^§^
*p* < 0.05 rMnSOD treated and irradiated samples versus irradiated samples; (c) ^^^
*p* < 0.05 pretreated and 1.0 Gy irradiated samples versus the 1.0 Gy irradiated and rMnSOD treated samples. CTR, control mice; rMnSOD, mice treated with human recombinant manganese superoxide dismutase; 0.25 Gy, 0.5 Gy, and 1.0 Gy, mice exposed to increasing radiation doses; 0.25 Gy + rMnSOD, 0.5 Gy + rMnSOD, and 1.0 Gy + rMnSOD, mice exposed to increasing radiation doses and treated with rMnSOD (protective role of rMnSOD); rMnSOD + 1.0 Gy, mice pretreated with rMnSOD and exposed to 1.0 Gy radiation (preventive role of rMnSOD).

**Figure 4 ijms-21-03281-f004:**
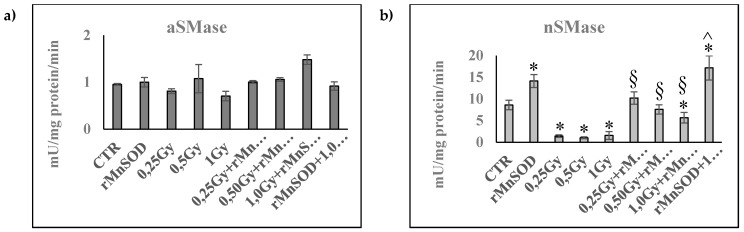
Effect of radiation and rMnSOD on aSMase and nSMase activity in the liver. aSMase and nSMase activity evaluated using an Amplex Red Sphingomyelinase assay kit, as reported in the “Materials and Methods” section. Livers from the mice treated with increasing doses of radiation with or without rMnSOS. (**a**) aSMase; (**b**) nSMase. Data are expressed as mU/mg protein/min and represent the mean+SD of three liver samples, each carried out in triplicate. Significance: (a) * *p* < 0.05 versus the control sample (CTR); (b) ^§^
*p* < 0.05 rMnSOD treated and irradiated samples versus the irradiated samples; (c) ^^^
*p* < 0.05 pretreated and 1.0 Gy irradiated sample versus the 1.0 Gy irradiated and rMnSOD treated sample. CTR, control mice; rMnSOD, mice treated with human recombinant manganese superoxide dismutase; 0.25 Gy, 0.5 Gy, and 1.0 Gy, mice exposed to increasing radiation doses; 0.25 Gy + rMnSOD, 0.5 Gy + rMnSOD, and 1.0 Gy + rMnSOD, mice exposed to increasing radiation doses and treated with rMnSOD (protective role of rMnSOD); rMnSOD + 1.0 Gy, mice pretreated with rMnSOD and exposed to 1.0 Gy radiation (preventive role of rMnSOD).

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
