# Peer review of "Acid and Neutral Sphingomyelinase Behavior in Radiation-Induced Liver Pyroptosis and in the Protective/Preventive Role of rMnSOD"

_ijms, 2020, doi:10.3390/ijms21093281_

Round 1

Reviewer 1 Report

The study by Cataldi et al investigates acid and neutral SMases in a model of radiation-induced liver pyroptosis and if rMnSOD is able to proect against liver pyroptosis through influencing the SMases. They observed that radiation induced caspase 1 and this was partially protected by rMnSOD. Notably, radation also expression of aSMase (with little effect on nSMase1) and this was reduced by rMnSOD treatment suggesting a possible role. However, comparable changes in either aSMase or nSMase1 protein were not observed. Similarly, no significant changes in aSMase activity were observed. In contrast, N-SMase activity was suppressed by radiation and this suppression was inhibited by rMnSOD. From these, the authors conclude that rMnSOD is protective of radiation induced pyroptosis – and this is because of its effects at inhibiting aSMase expression and enhancing N-SMase.

Overall, this manuscript is somewhat muddled and there are a number of issues that preclude publication. The authors may wish to consider the following.

Major

  1. The authors conclude that loss of N-SMase activity and increased aSMase are important for the pyroptosis seen. The data shown do not back this up at all as it lacks anything functionally linking the enzymes studied to the process. If they had studies in acid SMase KO mice or used inhibitors in their models and saw protection/exacerbation, this would lend credence to their conclusions. At best, this is a correlative and descriptive study. They make some effort to draw connections e.g. radiation induces acid smase that leads to lysosome rupture that leads to caspase 1 – but this is all speculation. They do not have time courses (which they note as a limitation) but, again, are lacking any loss of function approaches to show this. There may also be other means to examine this by immunoblot (alterations in cathepsin levels etc).
  2. Related to point one, this may be a correlative study – but there is an additional problem that even their own data do not seem to correlate. They see changes in acid SMase expression but not in activity nor consistent changes in protein, they see changes in N-SMase activity but nothing consistent in nSMase1 protein or expression. It’s possible this reviewer is missing something but possible explanations for these discrepanices are not clear from the discussion
  3. Could changes in N-Smase activity be due to other N-SMases – they emphasize in the introduction that there are multiple isoforms but do not investigate expression of any others. This seems to be important given the expression of nSMase1 is low (according to the authors)
  4. Unclear of the efficacy of the antibodies used. They need to show positive and negative controls that they antibodies are effective – before conclusions can be drawn. In this reviewer’s experience, antibodies against aSMase sometimes recognize the pro form and not the mature enzyme (which could account for some discrepancies they see). If the antibodies are not effective, conclusions on the protein levels are incorrect.
  5. Related to this, it’s difficult to make conclusions that SLs are involved in this process when lipid levels have not been measured.

Minor

  1. The authors do not give adequate context as to why the study of radiation-induced liver pyroptosis is important, and what is the purpose of studying SM metabolism and SMases in this process.
  2. Putting all data in one panel makes it very hard to discern changes (or lack thereof). This is particularly evident for SMPD2 expression in Figure 2 and aSMase activity in Figure 4.
  3. The data on SMase activity are interesting – but run counter to what has been previously shown i.e. that ROS inhibit SMase activity. Do the authors think that there is a specific free radical that may inhibit nSMase? This would seem important for discussion.
  4. There are some formatting and language issues with the manuscript. For example, at the end of the introduction, it looks like there is a section from ‘manuscript guidelines’ detailing how the results should be organized.

Author Response

Dear Reviewer,

thank you very much for your revision and for your comments that really improved the manuscript

Major 

  1. The authors conclude that loss of N-SMase activity and increased aSMase are important for the pyroptosis seen. The data shown do not back this up at all as it lacks anything functionally linking the enzymes studied to the process. If they had studies in acid SMase KO mice or used inhibitors in their models and saw protection/exacerbation, this would lend credence to their conclusions. At best, this is a correlative and descriptive study. They make some effort to draw connections e.g. radiation induces acid smase that leads to lysosome rupture that leads to caspase 1 – but this is all speculation. They do not have time courses (which they note as a limitation) but, again, are lacking any loss of function approaches to show this. There may also be other means to examine this by immunoblot (alterations in cathepsin levels etc).

Thank you very much for this observation because it highlights that the purpose of the work is unclear. This is not a classic experiment conducted in your laboratory and therefore all possible tests can be made. Of course, this is an observational study but an experiment unique in the world. The experimental model required the strength of numerous researchers who from Italy went to work in Dubna and managed to bring safe samples to Italy with an infinite series of bureaucratic permits. The experiment belongs to a large project in which several Italian university centers participated and therefore, on the return, the mice organs were distributed upon return. The experiment was very expensive and the organs themselves were divided. In particular, only one lobe of each liver has reached us. This is the reason why the study is observational, but it is a unique observation in the world that could be useful to the scientific community for future work. This has been explained in the paper, at the end of Introduction.

  1. Related to point one, this may be a correlative study – but there is an additional problem that even their own data do not seem to correlate. They see changes in acid SMase expression but not in activity nor consistent changes in protein, they see changes in N-SMase activity but nothing consistent in nSMase1 protein or expression. It’s possible this reviewer is missing something but possible explanations for these discrepanices are not clear from the discussion

really thank you for this suggestion because it was the missing part to explain our thinking. It has been included at the end of the “discussion” section

  1. Could changes in N-Smase activity be due to other N-SMases – they emphasize in the introduction that there are multiple isoforms but do not investigate expression of any others. This seems to be important given the expression of nSMase1 is low (according to the authors)

you are right but we had a low amount of sample and each experiment was repeated several times. We have no other proteins and, in any case, in this difficult period for everyone, the laboratories are closed

  1. Unclear of the efficacy of the antibodies used. They need to show positive and negative controls that they antibodies are effective – before conclusions can be drawn. In this reviewer’s experience, antibodies against aSMase sometimes recognize the pro form and not the mature enzyme (which could account for some discrepancies they see). If the antibodies are not effective, conclusions on the protein levels are incorrect.

we cannot repeat the experiments because, as mentioned above, we have no other samples and the laboratories are closed. In any case, we have been working on acid and neutral sphingomyelinases for 30 years and have numerous blots of positive and negative controls with the various types of antibodies and, therefore, also of those used in the present experiment. If you can add the ones we have

  1. Related to this, it’s difficult to make conclusions that SLs are involved in this process when lipid levels have not been measured.

we thought at this point because we are experts in lipidomics but the amount of material was too small to do everything

Minor

  1. The authors do not give adequate context as to why the study of radiation-induced liver pyroptosis is important, and what is the purpose of studying SM metabolism and SMases in this process.

These point have been explained (p2 lines 72; p3 lines 99-101).

  1. Putting all data in one panel makes it very hard to discern changes (or lack thereof). This is particularly evident for SMPD2 expression in Figure 2 and aSMase activity in Figure 4.

Figure 2 and figure 4 have been changed

  1. The data on SMase activity are interesting – but run counter to what has been previously shown i.e. that ROS inhibit SMase activity. Do the authors think that there is a specific free radical that may inhibit nSMase? This would seem important for discussion.

       It has been discussed (p9 lines 302-307)

  1. There are some formatting and language issues with the manuscript. For example, at the end of the introduction, it looks like there is a section from ‘manuscript guidelines’ detailing how the results should be organized.

They have been corrected

Reviewer 2 Report

The ms by Cataldi et al reports the Human Recombinant Mn Superoxide Dismutase (rMnSOD) protective effect versus radiation-induced pyroptosis in rat liver.

The issue is interesting and the experimental design is well-enough designed and performed.

The major concern is related to the data reporting alteration in the gene expression, protein level and activity of two Sphingomyelinases, namely aSMase and nSMase1.

It is very curious the finding that irradiation is able to increase aSMase gene expression concurrently with a decrease in the protein level. Moreover, the authors find that the aSMase activity appears unaffected by irradiation. How do they explain a decrease in the protein and a maintenance of the enzyme activity?

Unfortunately, the ms does not report the amount of the product of SMase activity, namely ceramide, that is likely responsible of the effects of irradiation. If the Authors have no access to a HPLC/MS facilty, simple chormatographic procedures are available to measure ceramide in the rat liver.

Concerning the rMnSOD, how do the Authors justify the dramatic effect on the 48kDa form of nSMase, when administered alone, and not when administered before the irradiation procedure (Fig.3 lanes rMnSOD and rMnSOD-1.0Gy). One should expect to find the same effect in both treatments.

On lane 268 of the Discussion, the Authors state that "rMnSOD was not able to act significantly at a nuclear level but mainly on the plasmamembrane nSMAse". This statement appears to contraidct the finding of a rMnSOD protective effect on aSMase gene expression.

Minor comments

  • Check spelling of Caspase-1. In the text it is not homegenous (caspase1, Caspase1, Caspase-1)
  • In the y-axis of figures, please use the term Percent, and not Percentage
  • English grammar and style should be improved

Author Response

Dear Reviewer,

thank you very much for your revision and for your comments that really improved the manuscript

Major

  1. The major concern is related to the data reporting alteration in the gene expression, protein level and activity of two Sphingomyelinases, namely aSMase and nSMase1.

It is very curious the finding that irradiation is able to increase aSMase gene expression concurrently with a decrease in the protein level. Moreover, the authors find that the aSMase activity appears unaffected by irradiation. How do they explain a decrease in the protein and a maintenance of the enzyme activity?

It has been discussed (p 9 lines 308-318)

  1. Unfortunately, the ms does not report the amount of the product of SMase activity, namely ceramide, that is likely responsible of the effects of irradiation. If the Authors have no access to a HPLC/MS facilty, simple chormatographic procedures are available to measure ceramide in the rat liver.

you are right but the experimental model required the strength of numerous researchers who from Italy went to work in Dubna and managed to bring safe samples to Italy with an infinite series of bureaucratic permits. The experiment belongs to a large project in which several Italian university centers participated and therefore, on the return, the mice organs were distributed upon return. The experiment was very expensive and the organs themselves were divided. In particular, only one lobe of each liver has reached us. We thought at this point because we are experts in lipidomics but the amount of material was too small to do everything

  1. Concerning the rMnSOD, how do the Authors justify the dramatic effect on the 48kDa form of nSMase, when administered alone, and not when administered before the irradiation procedure (Fig.3 lanes rMnSOD and rMnSOD-1.0Gy). One should expect to find the same effect in both treatments.

you're right, it was surprising for us too, but immunoblotting was repeated several times always with the same result. Certainly, future experiments will be needed to clarify this point

  1. On lane 268 of the Discussion, the Authors state that "rMnSOD was not able to act significantly at a nuclear level but mainly on the plasmamembrane nSMAse". This statement appears to contraidct the finding of a rMnSOD protective effect on aSMase gene expression.

the statement refers to nuclear neutral sphingomyelinase and not to lysosomal acid sphingomyelinase

Minor

  1. Check spelling of Caspase-1. In the text it is not homegenous (caspase1, Caspase1, Caspase-1)

Thank you very much, it has been changed

  1. In the y-axis of figures, please use the term Percent, and not Percentage

    y-axis has been changed

  1. English grammar and style should be improved

It has been done 

Round 2

Reviewer 1 Report

In this resubmitted manuscript, the authors have made some attempt to address the concerns raised. The reviewer certainly understands the nature of material limitation for repeat experiments - which is reasonable. The reviewer also understands the current circumstances as hindering the ability to perform additional experiments or analyses. Nonetheless, taking these circumstances into consideration, a stronger emphasis has to be placed on appropriate discussion of results and minimizing overconclusions on the data they have. The authors may wish to consider the following.

  1. While they have added the ‘observational’ lines to their introduction as a qualifier, the authors maintain in the discussion that alterations in aSMase and nSMase activity/mRNA/protein mean that they are functionally involved in radiation-induced pyroptosis by the liver/protection by rnMnSOD involves effects on these enzymes. This remains an overconclusion – all they can say is that alterations in SMases may be involved (but even then, their own data do not fully support that)
  2. The authors misunderstood the reviewer’s issue with figures 2 and 4. The differences in expression/activity between SMPD1/SMPD2 (Fig 2) and aSMase/nSMase (Fig 4) are the issue – not putting the radiation alone and rnMnSOD/radiation together. Also seems that figure 3 is duplicated?
  3. While the authors have provided additional discussion, they have not discussed their different results of ROS inhibiting nSMase vs activating in other systems.
  4. As an additional note to the authors, while they may have used  antibodies before, it does not mean that they are going to work identically in every cell line and every tissue, and it should not be on the reader to go literature hunting to confirm that the antibodies are recognizing the correct protein. The reviewer understands the circumstances, so can let it pass this time, but the authors should consider this in the future and at least include a positive control on the same immunoblot, as well as indicate molecular weight markers.

Author Response

  1. While they have added the ‘observational’ lines to their introduction as a qualifier, the authors maintain in the discussion that alterations in aSMase and nSMase activity/mRNA/protein mean that they are functionally involved in radiation-induced pyroptosis by the liver/protection by rnMnSOD involves effects on these enzymes. This remains an overconclusion – all they can say is that alterations in SMases may be involved (but even then, their own data do not fully support that)

thanks, according to your observation, the conclusion has been revised (p.9 lines 321-324)

  1. The authors misunderstood the reviewer’s issue with figures 2 and 4. The differences in expression/activity between SMPD1/SMPD2 (Fig 2) and aSMase/nSMase (Fig 4) are the issue – not

putting the radiation alone and rnMnSOD/radiation together. Also seems that figure 3 is duplicated?

I'm sorry, I didn't understand, new 2 and 4 figures have been included. Fig. 3 is not a duplicate, reviewer 2 advised to replace "percentage" with "percent" in b)

  1. While the authors have provided additional discussion, they have not discussed their different results of ROS inhibiting nSMase vs activating in other systems.

ROS are produced by radiation and therefore we have entered the discussion on this (p9 lines-296-301). In addition, we added “There are no data in literature to be able to make a comparison between the amount of ROS production in response to the different types of radiation in the liver. Therefore it is really difficult to establish whether the ROS-SMase relationship can also be dependent on tissue specificity and / or radiation dose” (p9 lines 301-305).

  1. As an additional note to the authors, while they may have used  antibodies before, it does not mean that they are going to work identically in every cell line and every tissue, and it should not be on the reader to go literature hunting to confirm that the antibodies are recognizing the correct protein. The reviewer understands the circumstances, so can let it pass this time, but the authors should consider this in the future and at least include a positive control on the same immunoblot, as well as indicate molecular weight markers.

I really appreciate your advice. Thank you very much

Reviewer 2 Report

The authors have replied to the reviewer's comments. Thus the paper is acceptable in the present form.

Author Response

Thank you very much!